# Differentiable Learning of Graph-like Logical Rules from Knowledge Graphs

## Abstract

Logical rules inside a knowledge graph (KG) are essential for reasoning, logical inference, and rule mining. However, existing works can only handle simple, i.e., chain-like and tree-like, rules and cannot capture KG's complex semantics, which can be better captured by graph-like rules. Besides, learning graph-like rules is very difficult because the graph structure exhibits a huge discrete search space. To address these issues, observing that the plausibility of logical rules can be explained by how frequently it appears in a KG, we propose a score function that represents graph-like rules with learnable parameters. The score also helps relax the discrete space into a continuous one and can be uniformly transformed into matrix form by the Einstein summation convention. Thus, it allows us to learn graph-like rules in an efficient, differentiable, and end-to-end training manner by optimizing the normalized score. We conduct extensive experiments on real-world datasets to show that our method outperforms previous works due to logical rules' better expressive ability. Furthermore, we demonstrate that our method can learn high-quality and interpretable graph-like logical rules.

## 1 Introduction

Knowledge graph (KG) refers to a special type of directed graphs including various entities as nodes and relations as directed edges representing a large number of facts (Auer et al., 2007; Bollacker et al., 2008). In KG, logical rules are a set of compositional logical relations within a specific structure, which are important for reasoning (Cohen et al., 2019; Zhang et al., 2019a; Qu & Tang, 2019), logical inference (Dhingra et al., 2020; Das et al., 2018; Xiong et al., 2017), rule mining (Sadeghian et al., 2019; Yang et al., 2017; Yang & Song, 2020), theorem proving (Rocktäschel & Riedel, 2017; Minervini et al., 2018; 2020), etc.

Learning logical rules (Galárraga et al., 2015; Chen et al., 2016), as an important task, aims to infer a structural logical rule for logical query or relation, which can support logical query or link prediction while providing interpretable logical rules. The structure of logical queries can be various with very different semantics, as shown in Figure 1, including chain-like, tree-like and graph-like rules. Learning the logical rules, especially the graph-like rules, are very difficult because both the logical structure and the relations assigned on each edge are unknown requiring to be inferred from input-output pairs, which compose a huge discrete searching space.

In this paper, we dive into the problem of learning graph-like logical rules, including both the logical structure representing how logic connects and the relations assigned on different edges. Recently, a series of works on learning logical rule (Yang et al., 2017; Sadeghian et al., 2019; Yang & Song, 2020) has been proposed, which not only can support tasks including logical query and link prediction, but as a side effect, can also provide the mined logical rules with high interpretability. As shown in Figure 1, all these works are limited to learning chain-like rules (the left case) (Yang et al., 2017; Sadeghian et al., 2019) or tree-like rules (the middle case) (Hamilton et al., 2018; Ren et al., 2020; Yang & Song, 2020). However, there are widely-existed graph-like logical rules, which the existing works cannot handle due to their limited expressive ability about logical rules. Learning graph-like logical rules is very important in many scenarios such as recommendation systems, question-answering system and KG completion, while learning such complex rules is still an open and challenging problem.

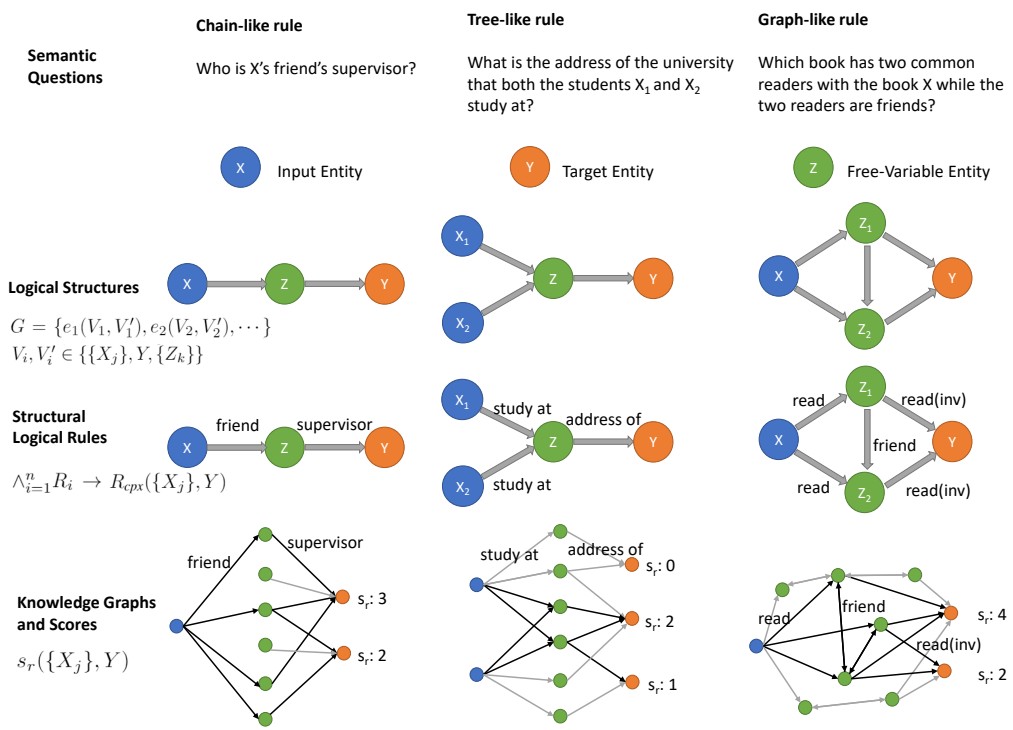

Figure 1: Three examples of chain-like, tree-like, graph-like rules (three columns) and their corresponding semantic questions, logical structures, structural logical rules, KGs and scores (four rows).

We propose a novel method that can explicitly learn the structural logical rules, including a logical structure and the relations assigned on each edge, and we can use the inferred logical rules for conducting inductive logical query with unseen entities and graphs. All the structural logical rules construct a discrete search space to explore, and searching for that is an NP-hard problem. To tackle with this problem, our method constructs a continuous space including both the structural information and the relational information to learn, which allows us to train our model in an end-to-end differentiable manner. Specifically, as shown in Figure 1, we take the frequency of a logical rule in KG as its score to estimate how likely a logical rule stands. After optimizing w.r.t. the normalized score, our model yields interpretable logical rules of high quality, and support inductive logical query and link prediction, which has been demonstrated by our extensive experiments on real-world datasets.

Our contributions can be summarized as following three aspects,

- We first propose the problem of learning graph-like rules and design an end-to-end differentiable model that can learn graph-like logical rules instead of only chain-like or tree-like rules, modeling both the logical structure describing how the logic connects and relations assigned on edges.
- We provide a uniform expression by Einsum to represent the score of all graph-like logical rules, including the ones that cannot be represented by a combination of matrix/element-wise addition/product, which is elegant for expression and convenient for implementation.
- We conduct extensive experiments to demonstrate that our model has better expressive ability for graph-like logical rules and show our model can mine high-quality logical rules with high interpretability.

## 2 PROBLEM FORMULATION

Here, we formally introduce the definition of logical score, and based on that, we further introduce our model's main focus, relation inference (Yang et al., 2017; Sadeghian et al., 2019) and structural rule learning, and our evaluation task, logical query (Hamilton et al., 2018; Ren et al., 2020).

**Definition 1 (Logical Score)** *Logical rule is formulated by $\wedge_{i=1}^n R_i \rightarrow R_{cpx} : s_r$ where $s_r$ is the score for $\wedge_{i=1}^n R_i$, and $R_i$ is a relation $R_i = R_i(V_i, V_i'), V_i, V_i' \in \{\{X_j\}, Y, \{Z_k\}\}$ for $i = 1, \cdots, n$ and $R_{cpx}$ is a relation $R_{cpx}(\{X_j\}, Y)$, $\{X_j\}$ are input nodes, $\{Z_k\}$ are free-variable nodes, $Y$ is output node.*

For strict logical query, for any $R_{\mathrm{cpx}}(\{X_j\}, Y)$, there exists $(Z_1, \cdots, Z_K)$ that make $\wedge_{i=1}^n R_i$ be true, we can draw the conclusion $\wedge_{i=1}^n R_i \rightarrow R_{\mathrm{cpx}}$. However, because KG is usually noisy and incomplete, for learning logical rules, our key insight is to design the score as the number of free-variable tuples $(Z_1, \cdots, Z_K)$ that make $\wedge_{i=1}^n R_i$ be true, which can capture the correlation between logical rules and the input-output pairs of a logical query. For example, for the case in the middle of Figure 1, $R_{\mathrm{study\ at}}(X_1, Z) \wedge R_{\mathrm{study\ at}}(X_2, Z) \wedge R_{\mathrm{address\ of}}(Z, Y) \rightarrow R_{\mathrm{cpx}}(X_1, X_2, Y)$; for the case in the right of Figure 1, we have $R_{\mathrm{read}}(X, Z_1) \wedge R_{\mathrm{read}}(X, Z_2) \wedge R_{\mathrm{friend}}(Z_1, Z_2) \wedge R_{\mathrm{read(inv)}}(Z_1, Y) \wedge R_{\mathrm{read(inv)}}(Z_2, Y) \rightarrow R_{\mathrm{cpx}}(X, Y)$. Note that, $R_{\mathrm{cpx}}$ can both be a relation that exists in the KG and the human-defined logic rule for a query, which tends to be more complex. The score $s_r$ serves as two roles: (i). when input-output pairs are given, it can measure how likely a logical rule is, which corresponds to the scenario of Task 1 and Task 2; (ii). when logical rules for query and inputs are given, it can measure how much a output node fits the query, which corresponds to Task 3.

**Task 1 (Relation Inference)** *Given $R_{cpx}(\{X_j\}, Y)$ is satisfied and a logical structure composed by $G = \{e_1(V_1, V_1'), e_2(V_2, V_2'), \cdots\}$, we need to infer how to assign relation $R_i$ on each edge $e_i$ to form a logical rule $\wedge_{i=1}^n R_i(V_i, V_i')$ that will make the score $s_r$ of $R_{cpx}(\{X_j\}, Y)$ high.*

For this task, previous relation inference works (Yang et al., 2017; Sadeghian et al., 2019) can also conduct this task but they limit the $G$ to be a chain-like. We model the relation between the input-output pairs behind the query as $R_{\mathrm{cpx}}$ and infer its graph-like logical rule.

**Task 2 (Structural Rule Learning)** *Given $R_{cpx}(\{X_j\}, Y)$ is satisfied and the possible max nodes number $\hat{n} \geq n_e$, where $n_e$ is the size of $\{\{X_j\}, Y\}$, we need to infer what structure $G = \{e_1(V_1, V_1'), e_2(V_2, V_2'), \cdots, e_n(V_n, V_n')\}$ where $n_e \leq n \leq \hat{n}$ and the relations assigned on edges $\wedge_{i=1}^n R_i$ that will make the score $s_r$ high.*

For this task, logical structures in previous works (Yang et al., 2017; Sadeghian et al., 2019; Yang & Song, 2020) are limited to chains or trees, and the number of input entities are limited to $1$. However, we can infer both the logical structure and the relations assigned on edges for graph-like rules.

**Task 3 (Logical Query)** *Given input nodes $\{X_j\}$ and the query relation, the target nodes of this query can be represented by $q = \{Y | R_{cpx}(\{X_j\}, Y)\}$.*

Note that, in previous works (Hamilton et al., 2018; Ren et al., 2020), the logical rule $R_{\mathrm{cpx}} = \wedge_{i=1}^n R_i$ is given, different from those works, we need to infer the $\wedge_{i=1}^n R_i$ for logic query. Our model targets at the inference of complex logical rules, and use the inferred logic rules to conduct logical query as evaluation task. For evaluation, we regard Task 3 as the main task and the other two tasks as side products.

# 3 RELATED WORKS

## 3.1 LOGICAL QUERY FROM KNOWLEDGE GRAPHS

Logical rules learning (Teru et al., 2020; Evans & Grefenstette, 2018; Manhaeve et al., 2018; Wang et al., 2019; Ho et al., 2018) is to learn logical rules (Task 1) for logical query (Task 3) in an inductive setting. Neural-LP (Yang et al., 2017) design an end-to-end differentiable framework to learn the probability of different logical rules. Furthermore, DRUM (Sadeghian et al., 2019) improve Neural-LP (Yang et al., 2017) by introducing the low-rank matrix decomposition. However, these two works can only tackle chain-like logical rules. Different from our model, they mainly focus on relatively simple logical rules such as chain-like or tree-like rules. To the best of our knowledge, our model is the first one that can learn to infer graph-like complex logical rule including structure and relations assigned on different edges.

Logical queries (Serge et al., 1995) aims to learn how to accurately query an entity (Task 3) according to given input entities and relations representing the logical rules in a transductive setting.

Table 1: Comparison between our method and existing methods.

| Methods | | GQE | Q2B | GraIL | NeuralLP | DRUM | Ours |
|---|---|---|---|---|---|---|---|
| $G$'s structure | | tree | tree | chain | chain | chain | graph |
| **Variable** | $\#\{X_j\}$ | multiple | multiple | single | single | single | multiple |
| | $Z$'s position | middle | middle | middle | middle | middle | arbitrary |
| | $Y$'s position | end | end | end | end | end | arbitrary |
| **Tasks** | | 3 | 3 | 3 | 1,3 | 1,3 | 1,2,3 |
| **Inductive** | | $\times$ | $\times$ | $\sqrt{}$ | $\sqrt{}$ | $\sqrt{}$ | $\sqrt{}$ |
| **Interpretable** | | $\times$ | $\times$ | $\times$ | $\sqrt{}$ | $\sqrt{}$ | $\sqrt{}$ |

According to Task 3, the logic rules representing the semantics of query explicitly given at both training and testing stages in this branch of works, but in our paper, the logic rules require to be inferred in the training stage. For most of these works, the main idea is to project entities into the embedding space (Bordes et al., 2013; Trouillon et al., 2016; Sun et al., 2018; Balažević et al., 2019) and transform the relations into a type of manipulation in embedding space, such as a linear projection. Hamilton et al. (2018) first proposes an embedding-based method for conduct query with tree-like logical rules. Ren et al. (2020) further improves Hamilton et al. (2018) by modeling the entities as box embedding rather than vector embedding, which is more natural for the manipulation for the conjunction of sets. Different from our model, these methods require explicit given logical structures with given relations on edges.

## 3.2 DIFFERENTIABLE INDUCTIVE LOGIC PROGRAMMING

Inductive Logic Programming (ILP) (Shapiro, 1981) aims to conduct inductive logic reasoning or theorem proving based on entities, predicates and formulas. Predicates refer to a type of function projecting one or more entities to $0$ or $1$. For example, $isMale(X)$ return whether $X$ is a person (1) or not (0), $isFatherOf(Y, X)$ returns whether $Y$ is the father of $X$ (1) or not (0), where $X$ and $Y$ are variables which we can instantiate them as entities. Then, we can define formulas by combining predicates by logic operations including and/or/not. For examples, $isMale(X) \wedge isFatherOf(Y, X) \rightarrow isSonOf(X, Y)$ is a logic entailment, which composes of a body formula $isMale(X) \wedge isFatherOf(Y, X)$ and a head formula $isSonOf(X, Y)$. The physical meaning of this entailment means if the body formula is satisfied (equals to 1), then we can draw the conclusion in the head formula. ILP is an NP-hard problem (Zhang et al., 2019b) and traditional methods relying on hard matching (Galárraga et al., 2015) for ILP are of high computational complexity due to the very large search space.

Markov Logic Networks(MLN) (Richardson & Domingos, 2006; Kok & Domingos, 2005) elegantly combine ILP problem and probabilistic models, which define potential functions in Markov random field with formulas as nodes. Different from that, we consider the logical rule as a graph with entities as nodes. In recent years, a lot of works on differentiable ILP (Rocktäschel & Riedel, 2017; Minervini et al., 2018; 2020) have been proposed. The most recent work is NLIL (Yang & Song, 2020), which targets learning the logic rules based on unary and binary predicates efficiently. When we only focus on the binary predicate, then we can naturally conduct ILP based on KG because the relations in KG can be naturally regarded as a binary predicate on two entities (exist: 1; not exist: 0). If we only use NLIL (Yang & Song, 2020) for binary predicates (relations in KG), then it can only tackle chain-like rules (same as Neural-LP (Yang et al., 2017) and DRUM (Sadeghian et al., 2019)) instead of graph-like rules. To the best of our knowledge, differentiable ILP methods cannot handle graph-like rules.

## 4 THE PROPOSED METHOD

Logical graph structure and relations assigned on edges construct a huge discrete space, which is extremely hard to search (Task 1 and Task 2). The key idea of our method is to design a score function modeling both the logical structure and the relation assigned on edges, by optimizing which we can obtain complex rules and conduct logical query (Task 3).

- First, we introduce how to represent the score $s_r$ in when given the logical structure, to estimate how likely a logical rule is, by maximizing which we can infer relations (Task 1).
- Second, we introduce how to merge the structural information into the score to uniformly obtain the logical structure and relations on edges by optimizing one score (Task 2).
- Finally, we provide a uniform and elegant expression of the score $s_r$ in a matrix form by Einsum for all graph-like logical rules and exploit cross-entropy loss for optimizing our model, which can further support logical query (Task 3).

## 4.1 RELATION INFERENCE FOR GRAPH-LIKE LOGICAL RULES

Given input node $\{X_j\}$, target node $Y$ and a structure $G$ as shown in Figure 1, we use the number of the tuples $(\{Z_k\})$ that satisfy $\wedge_{i=1}^{n} R_i$ as the score to evaluate the plausibility of a logical rule. We denote the adjacency matrix of relation $R_i$ as $A_i$, which is a soft choice among adjacency matrices $\bar{A}_k$ corresponding to all relations $\mathcal{R}$ in KG. Then, the score, can be represented as

$$s_r(\{X_j\}, Y) = \sum_{Z_1=1}^{|\mathcal{V}|} \cdots \sum_{Z_K=1}^{|\mathcal{V}|} \prod_{i=1}^{n} A_i[V_i, V_i'], \tag{1}$$

where $V_i, V_i' \in \{\{X_j\}, Y, \{Z_k\}\}$, $A_i[V_i, V_i']$ denotes the $(V_i, V_i')$-indexed entry in $A_i$, $A_i$ is defined as

$$A_i = \sum_{k=1}^{|\mathcal{R}|} \bar{A}_k \cdot \exp(\beta_{ik}) / \sum_{k'=1}^{|\mathcal{R}|} \exp(\beta_{ik'}), \tag{2}$$

where $\bar{A}_k$ represents the $k$-th relation in the set of relations in KG, by which we can use the coefficiencts $\{\beta_{ik}\}$ on different relations as learnable parameters to learn which relation should be assigned on each edge given the logical structure. For the right case in Figure 1, Eq. (1) becomes

$$s_r(\{X_j\}, Y) = \sum_{Z_1=1}^{|\mathcal{V}|} \sum_{Z_2=1}^{|\mathcal{V}|} A_1[X, Z_1] A_2[X, Z_2] A_3[Z_1, Z_2] A_4[Z_1, Y] A_5[Z_2, Y], \tag{3}$$

Intuitively, in Eq. (3), after assigning an entity for each free variable $Z$, $A_1[X, Z_1] = 1$ means there is relation $R_1$ between $X$ and $Z_1$, the product of such terms equals to 1 only if all of them equal to 1, which means all free variable $Z_1$ and $Z_2$ together with $X$ and $Y$ satisfies the logical rule $\wedge_{i=1}^{n} R_i$. We sum w.r.t. $Z_1, \ldots, Z_K$ to calculate the count of such tuples $(Z_1, \ldots, Z_K)$ in the full KG that satisfied the logical rule to measure its plausibility.

## 4.2 STRUCTURAL RULE LEARNING

In realistic logical inference scenarios (Yang & Song, 2020; Sadeghian et al., 2019), besides the relations, we also do not know the structure of the logical rule $G = \{e_1(V_1, V_1'), e_2(V_2, V_2'), \cdots\}$. Thus, we need to infer the logical structure as well as the relations. To achieve this goal, we add two special auxiliary relations: "removing" relation represented by full-one matrix $\mathbf{1}$ (all the entries are 1) and "merging" relation represented by identity matrix (the diagonal entries are 1 and the others are 0) into the adjacency matrices. The physical meaning of identity matrix $I$ is merging two connected nodes as one node in the logical rule, because if $I[V_i, V_i'] = 1$, then $V_i = V_i'$, i.e., they are the same entity in KG; the full-one matrix $\mathbf{1}$ removes the edge in the logical rule, because for any $V_i$ and $V_i'$, we have $\mathbf{1}[V_i, V_i'] = 1$, i.e., there is no relation requirement between $V_i$ and $V_i'$. We expand the parameters $\{\beta_{jk}\}$ for these two matrices and update the equation of softly selected adjacency matrix $A_i$ in Eq. (1) as follows,

$$A_i = \sum_{k=1}^{|\mathcal{R}|+2} \bar{A}_k^+ \cdot \exp(\beta_{ik}) / \sum_{k'=1}^{|\mathcal{R}|+2} \exp(\beta_{ik'}), \tag{4}$$

where $\bar{A}_k^+$ represents the $k$-th relation in the augmented set of relations including the original relations in KG and two auxiliary relations. By merging nodes and removing edges, which correspond to learning large coefficients on one of these two relations, we can obtain any graph structure from the complete graph whose nodes number is no less than the ground truth graph.

**Theorem 4.1** *Given adjacency matrices consisting of the original adjacency matrices and two auxiliary adjacency matrices: identity matrix $\mathbf{I}$ and full-one matrix $\mathbf{1}$, we assume there are $\hat{m} \leq m$ points constructing the complete graph, then for any logical rule $\wedge_{i=1}^{n} R_i$, there exists a suite of parameters $\{\beta_{jk}\}$ that make Eq. (1) equal to the number of $(Z_1, \ldots, Z_K)$ that satisfies the logical rule.*

### 4.3 TRAINING ALGORITHM

For chain-like and tree-like rules, Eq. (1) can be efficiently computed with matrix and element-wise product (see Section 4.4). But for more general graph-like rules, Eq. (1) cannot be directly computed in such a compact way. Here, we introduce Einsum (see Appendix A for more details) to make this possible. Einsum is a flexible convention of matrix calculation that unifies matrix multiplication, element-wise multiplication and some more complex matrix/tensor calculation that cannot be expressed by these ordinary operators (e.g., calculating the scores for graph-like rules). Specially, we express Eq. (2) in a Einsum format as follows,

$$s_r(\{X_j\}, Y) = \text{einsum}\left('X_1, ..., V_1 V_1', ..., V_n V_n', Y', v_{X_1}, \cdots, A_1, \ldots, A_n, v_Y\right), \qquad (5)$$

where $s_r(\{X_j\}, Y)$ denotes the score of the pair $(\{X_j\}, Y)$, $v_X$ and $v_Y$ are the one-hot vectors of the input and output entities.

Such a convention has two advantages: (i). we can uniformly represent all graph-like rules in a matrix form; (ii). well-engineered libraries such as Numpy[1] and Pytorch[2] can be exploited for fast computation (Daniel et al., 2018).

Finally, we need a loss function that can not only encourage the positive samples but also can provide penalty for negative samples, which allows our model to be learn logical rules accurately. Thus, we first calculate $\hat{s}_r(\{X_j\}, Y) = s_r(\{X_j\}, Y)/\sum_{Y' \in \mathcal{V}} s_r(\{X_j\}, Y')$, i.e., the normalized score for each entity, then optimize the objective function composed by cross-entropy loss as follows,

$$\arg\min_{\{\beta_{jk}\}} \sum_{(\{X_j\}, Y) \in \mathcal{D}} \sum_{Y' \in \mathcal{V}} -\mathbb{I}[Y' = Y] \log(\hat{s}_r(\{X_j\}, Y')), \qquad (6)$$

where $\mathbb{I}[\cdot]$ is an indicator function, which returns 1 if the statement is true and otherwise returns 0.

We optimize this loss, which is back-propagated through the calculation of Einsum, to learn the parameters $\{\beta_{jk}\}$ with high interpretability in an end-to-end differentiable manner. The training process is summarized in Algorithm 1.

---

**Algorithm 1 The training process of our model for logical inference**.

---

**Require:** A set $\mathcal{D}$ of training data $\{(\{X_j\}, R_{\text{cpx}}, Y)\}$, KG, max nodes number $\hat{n} \geq n_e$, max_step;
1: initialize $\beta_{ik}, i = 1, \ldots, n, k = 1, \ldots, |\mathcal{R}|$, step = 0;
2: **while** step < max_step **do**
3:     sample a mini-batch $\mathcal{D}_{\text{batch}}$ batch $\subseteq \mathcal{D}$;
4:     **for** each $(\{X_j\}, R_{\text{cpx}}, Y) \in \mathcal{S}_{\text{batch}}$ **do**
5:         update parameters $\{\beta_{ik}\}$ based on the loss function Eq. (6);
6:     **end for**
7:     step $\leftarrow$ step+1
8: **end while**
9: **return** logical rules $\wedge_{i=1}^n R_i, n_e \leq n \leq \hat{n}$.

---

**Computational Complexity.** Naturally, the search space of structural logical rules is very large and searching them is an NP-hard problem (Galárraga et al., 2015). Our method constructs a continuous space to estimate the logical rules to optimize it in a differentiable manner, which significantly reduces the complexity to $O(|\mathcal{V}|^{K+1})$, where $|\mathcal{V}|$ is the number of entities in KG, $K$ is the number of free-variable nodes.

### 4.4 CASE STUDY: COMPARISON WITH EXISTING METHODS

We take three real-world cases in Figure 1 to further show how our graph-like rules learning method generalizes previous chain-/tree-like one. To the best of our knowledge, our model is the first that can infer the graph-like rules.

- *Chain-like rule.* Left in Figure 1 can be represented by $s_r(\{X_j\}, Y) = \text{einsum}('i, jk, kl, l', v_X, A_{\text{friend}}, A_{\text{supervisor}}, v_Y)$. This will degenerate to the form of matrix multiplication (Yang et al.,

---

[1]https://numpy.org/doc/stable/reference/generated/numpy.einsum.html
[2]https://pytorch.org/docs/stable/generated/torch.einsum.html

2017; Sadeghian et al., 2019) as follows, $s_r(\{X_j\}, Y) = v_Y^\top A_{\text{supervisor}} A_{\text{friend}} v_X$, where $v_X$ and $v_Y$ are the one-hot vectors of the input entity and output entity.

- *Tree-like rule.* Middle in Figure 1 can be represented by $s_r(\{X_j\}, Y) = \text{einsum}( \text{'i, j, ik, jk, kl, l'},$ $v_{X_1}, v_{X_2}, A_{\text{study at}}, A_{\text{study at}}, A_{\text{address of}}, v_Y)$. This will degenerate to the form of a combination of matrix multiplication and element-wise product. For example, the case in the middle of Figure 1 as follows, $s_r(\{X_j\}, Y) = v_Y^\top A_{\text{address of}}(A_{\text{study at}} v_{X_2} \odot A_{\text{study at}} v_{X_1})$, where $\odot$ denotes the element-wise product, $v_{X_1}$ and $v_{X_2}$ are one-hot vectors of two inputs, and $v_Y$ is the one-hot vector of the tail. EQB (Hamilton et al., 2018) and Query2box (Ren et al., 2020) require $Y$ to be in the end of the logical rules and $\{Z_k\}$ in the middle of logical rules, because they transform the logical rules into a computational flow. Our method does not have requirements on the number of input nodes and the position of $Y$ and $\{Z_k\}$.

- *Graph-like rule.* Right in Figure 1 can be represented by $s_r(\{X_j\}, Y) = \text{einsum}( \text{'i, ij, ik, jk, jl,}$ $\text{kl, l '}, v_X, A_{\text{read}}, A_{\text{read}}, A_{\text{friend}}, A_{\text{read (inv)}}, A_{\text{read (inv)}}, v_Y)$. This cannot be simplified by only using a combination of matrix multiplication/element-wise addition/product. All the graph-like logical rules can be expressed by Einsum uniformly.

## 5 EXPERIMENTS

We conduct extensive experiments on real-world datasets to compare our performance on logical query (Task 3) in Section 5.2. Furthermore, we also demonstrate that our model is able to infer the relations (Task 1) and learn the structural logical rules (Task 2) with high quality in Section 5.3 and Section 5.4.

### 5.1 EXPERIMENT SETUP

We implement our model in Python using Pytorch libraryand optimize all the models by Adam optimizer (Kingma & Ba, 2015).

**Datasets.** We use the Kinship, Family and Unified Medical Language System (UMLS) datasets (Kok & Domingos, 2007) to evaluate our model's ability to learn some representative logical rules for logical queries. Furthermore, we use Douban and Yelp to evaluate our model's ability to learn complex graph-like logical rules for logical queries. We report more details in Section C.1.

**Query Generation**. We carefully design five representative query structures as shown in Figure 2 including the right example in Figure 1, which do exist in real-world datasets.

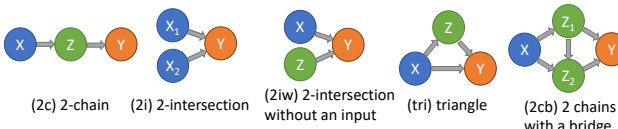

(2c) 2-chain    (2i) 2-intersection    (2iw) 2-intersection without an input    (tri) triangle    (2cb) 2 chains with a bridge

Figure 2: Five representative query structures.

We choose 2-chain (2c) because it is a basic chain rule, a comparison will help understand how our model works on the most common cases. We choose 2-intersection (2i) because it is a typical tree-like logical rule, while many of the existing works cannot handle it because most of them only allow one input or are limited to chain-like rules. Only complex logical query methods (Ren et al., 2020) can tackle with this case. We choose 2-intersection-without-an-input (2iw) because it is a special type of tree-like logical rules, the positions of the output and free-variable node are very special. To test more complex graph-like rule, we choose a triangle (tri) structure, which cannot be modeled as chain/tree-like rules. At last, we choose a structure of 2 chains with a bridge (2cb) the same as the right case in Figure 1. All these query structures have corresponding realistic semantics. For query structure 2c, 2i, 2iw, we extract the top 5 frequent query types ($\wedge_{i=1}^n R_i$) and then follow the logic rules to randomly generate 1000 input-output pairs for each query structure. For query structure tri, we use the query type represented by "Who is X's friend and meanwhile has a co-reading book with X?". For query structure 2cb, we use the query semantics "Which book has two common readers with the book X while the two readers are friends" as shown in Figure 1. We split the datasets (queries) into training, validation, testing datasets according to the ratio $2 : 1 : 1$. We use the training dataset for learning the parameters in our model, use the validation set to decide when to conduct early-stopping, and finally use the testing dataset for evaluating our model.

**Comparing Methods.** We compare our method on logical query with a series of rule mining methods Neural-LP (Yang et al., 2017) and DRUM(Sadeghian et al., 2019) with high interpretability and the state-of-the-art embedding-based logical query method (Ren et al., 2020) that can handle tree-like logical rules. To fairly compare the expressive ability about logical rules, we remove the embedding information and the corresponding neural network generating coefficients in rule mining baselines (Yang et al., 2017; Sadeghian et al., 2019), instead, we set the coefficients on relations as learnable parameters.

**Evaluation Metrics.** We use Mean Reciprocal Rank (MRR) and Hit Rate at k (k = 1, 3) as evaluation metrics (see Section C.2 for more details).

## 5.2 PERFORMANCE COMPARISON

Table 2: Performance comparison in terms of different query structures between different methods on three real-world datasets. The notation – means the method in the row cannot be used for the query structure in the column.

| Dataset | Method | 2c | | | 2i | | | 2iw | | |
|---|---|---|---|---|---|---|---|---|---|---|
| | | MRR | Hit Rate @3 | @1 | MRR | Hit Rate @3 | @1 | MRR | Hit Rate @3 | @1 |
| Kinship | Query2box | .23 | .22 | .20 | .25 | .17 | .09 | – | – | – |
| | Neural-LP | .38 | .38 | .26 | – | – | – | .43 | .61 | .19 |
| | DRUM | .55 | .59 | .44 | – | – | – | .33 | .45 | .15 |
| | Ours | **.63** | **.69** | **.52** | **.63** | **.81** | **.42** | **.53** | **.77** | **.24** |
| UMLS | Query2box | .13 | .08 | .07 | .24 | .28 | .10 | – | – | – |
| | Neural-LP | **.74** | **.82** | .61 | – | – | – | .28 | .38 | .06 |
| | DRUM | .66 | .65 | **.64** | – | – | – | .18 | .14 | .04 |
| | Ours | .66 | .71 | .58 | **.43** | **.53** | **.21** | **.32** | **.48** | **.12** |
| Family | Query2box | .41 | .56 | .21 | .23 | .26 | .00 | – | – | – |
| | Neural-LP | .73 | .86 | **.60** | – | – | – | .67 | .84 | .45 |
| | DRUM | **.75** | **.90** | **.60** | – | – | – | .65 | .87 | .41 |
| | Ours | .72 | .85 | .58 | **.67** | **.86** | **.47** | **.69** | **.88** | **.48** |

Table 3: Performance comparison of complex logical query on real-world datasets.

| Dataset | Method | tri | | | 2cb | | |
|---|---|---|---|---|---|---|---|
| | | MRR | Hit Rate @3 | @1 | MRR | Hit Rate @3 | @1 |
| Douban | Neural-LP | .91 | **1.00** | .83 | .61 | .65 | .49 |
| | DRUM | **1.00** | **1.00** | **1.00** | .61 | .65 | .49 |
| | Ours | **1.00** | **1.00** | **1.00** | **1.00** | **1.00** | **1.00** |
| Yelp | Neural-LP | .79 | .90 | .70 | .74 | .82 | .92 |
| | DRUM | .65 | .62 | .53 | .64 | .61 | .58 |
| | Ours | **.89** | **.85** | **.85** | **.94** | **.94** | **.94** |

We compare our model with other methods on three real-world datasets in terms of the three relatively simple but representative query structures (2c,2i,2iw) as shown in Table 2. Our method is the only one that can tackle all these three query structures. We can observe that, for query structure 2c, Our model achieves better or comparable performance with Neural-LP and DRUM, because these two methods are specifically designed for chain-like rules. However, they cannot handle query structure 2i because they only allow single-input query. Furthermore, they cannot learn the 2iw correctly, because they require the output entity to be in the end of chain rule. Their learned chain rules for 2iw are inaccurate or totally wrong, so our model improve the performance of 2iw query type with a large margin compared to them. Query2box is designed for handling missing relations in complex queries, which performs poorly on these datasets. Furthermore, Query2box relies on the embeddings of entities so it cannot handle unseen entities but ours can.

For more complex logic rules (tri,2cb), we conduct the experiment on two real-world datasets in recommendation system domains, Douban and Yelp, whose information is reported in Table 6. The

performances of Neural-LP and DRUM are poor because their learned chain rules are far from the the graph-like rules as shown in Figure 3. Here, we do not compare with Query2box because it cannot work on such graph-like logical rules.

The results reported in Table 3 show that our model has the best ability on learning such graph-like rules, which cannot be accurately modeled by methods for chain-like rules, such as Neural-LP or DRUM. As we will discuss in Section 5.3, our model learns totally correct logical rules so it can achieve such a good performance. The running time results are shown in Table 4, we can observe that our running time is comparable to Neural-LP and DRUM.

| Method | Kinship | UMLS | Family | Douban | Yelp |
|---|---|---|---|---|---|
| Neural-LP | .76 | .91 | 3.37 | .85 | .83 |
| DRUM | .23 | .43 | 1.65 | .71 | .71 |
| Ours | .99 | 1.08 | 6.76 | .76 | .72 |

Table 4: Running times (mins) of our model and other models on different datasets.

### 5.3 CASE STUDY

Furthermore, we check whether the learned rules are the same as ground-truth rules. As mentioned, we learn a set of weights represented by $\{\beta_{ik}\}$ for different relations assigned on edges, while the 'merging' relation corresponding to matrix $\mathbf{1}$ means merging two connected nodes and the 'removing' relation corresponding to matrix $\mathbf{I}$ means removing the edge (no rule requirement). So we visualize the weights representing the learned logic rules as shown in Figure 3. From that, we can observe that most of the relations are learned correctly with very high confidence, and it learns high confidence for removing one edge (the auxiliary relation represented by full-one matrix), which means our model has the ability to both infer the relations and learn the logic structures.

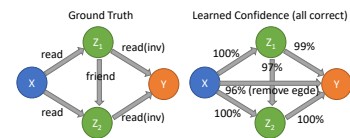

Figure 3: The graph-like logical rules learned by our model. We show the learned $k$-th relation ($k = \arg\max_k \beta_{ik}$) assigned on each $i$-th edge with the highest confidence $\exp(\beta_{ik})/\sum_{k'}\exp(\beta_{ik'})$. Note that, full-one matrix, i.e., $\mathbf{1}$, removes the corresponding edge from the logical rule.

### 5.4 ABLATION STUDY

To demonstrate the effectiveness of auxiliary matrices $\mathbf{I}$ and $\mathbf{1}$, we conduct the experiments that we train the model with both matrices, one of them or none of them, respectively.

As shown in Table 5, we can observe that for case 2c, the model with both matrices achieves the best performance, which suggests the effectiveness of these two matrices. For case 2i, the model with the matrix $\mathbf{1}$ achieves the best performance because the model with $\mathbf{1}$ has the expressive ability to model the case 2i, more matrices will lead to more parameters and difficulty for learning. Due to the same reason, for case 2iw, the model with the matrix $\mathbf{1}$ and the model with both matrices achieve similar performance. For most complex cases, we need both auxiliary matrices to accurately express the complex logical rules.

Table 5: Performance comparison of hit@10 among the variants of our method with or without matrices $\mathbf{I}$ and $\mathbf{1}$ on Family dataset.

| Variants | 2c | 2i | 2iw |
|---|---|---|---|
| none | 0.08 | 0.69 | 0.20 |
| only $\mathbf{1}$ | 0.43 | **0.94** | **0.58** |
| only $\mathbf{I}$ | 0.31 | 0.77 | 0.50 |
| both | **0.89** | 0.86 | **0.58** |

## 6 CONCLUSION

We propose a uniform score to not only unify the existing logical query and inference works but also tackle more complex graph-like logical rules. Furthermore, we exploit Einsum to elegantly express this score function and optimize our model in an end-to-end differentiable manner, which can learn both the logic structure and the relations assigned on edges. At last, we conduct extensive experiments on real-world datasets datasets to demonstrate the effectiveness of our model on logical query and show that our model can yield high-quality complex logical rules with interpretability.

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

## A  EINSUM

Einsum (Åhlander, 2002) is a mathematical notational convention that is elegant and convenient for expressing summation and multiplication on vectors, matrices, and tensors. It can represent some calculation of multiple matrices that cannot be represented by matrix product or element-wise product. Specifically, it not only makes the expression of complex logic rules very simple and elegant but also make the implementation of this calculation simpler and more efficient. Here, we introduce the implicit mode Einsum because it can tackle more cases than the classical one, and this function has been implemented in many widely-used libraries such as Numpy and Pytorch. It takes a string representing the equation and a series of tensors as input and provides a tensor as output. The rules of Einsum are as follows,

- the input string is comma-separated including a series of labels, where each term separated by commas represent a tensor (or matrix, vector) and each letter is corresponding to a dimension;
- the terms in the string before '→' means input variables, and the term after '→' means output variables;
- the same label in multiple inputs means element-wise multiplication on the dimensions of different input tensors;
- the dimensions that occur in the inputs but not in the outputs mean that dimension will be summed, while others will remain in the output. When there is no dimension after '→', it means all dimensions should be summed to get a scalar, and in such case '→' can be omitted.

Here we provide several simple examples to help understand:

- einsum ($'i, i \rightarrow ', a, b$) or einsum ($'i, i', a, b$) represents the inner product of two vectors $a^\top b$;
- einsum ($'i, j \rightarrow ij', a, b$) represents the outer product of two vectors $ab^\top$;
- einsum ($'ij, ij \rightarrow ij', A, B$) represents the element-wise product of two matrices $A \odot B$;
- einsum ($'ij, jk \rightarrow ik', A, B$) represents the matrix product of two matrices $AB$.

## B  PROOF OF THEOREM 4.1

**Proof 1** *Here, we provide a strategy to construct the coefficients $\{\beta_{jk}\}$ which make Eq. (1) equal to the number of $(Z_1, \ldots, Z_K)$ that satisfies the logical rule. First, without loss of generality, we assume that the ground-truth logical rules is $\wedge_{i=1}^n R_i(V_i, V_i')$ including $m$ free-variable nodes. Our strategy is divided into two stages: merging redundant nodes and removing useless edges.*

*Stage 1: Merging redundant nodes. If $\hat{m} > m$, we can set the coefficient of identity matrix of a complete graph on all edges between node $Z_m, Z_{m+1}, \cdots, Z_{\hat{m}}$ as 1, which means all of these edges are merged. And then for any $Z_j$ that $j < m$ and $Z_k, Z_l$ that $k, l > m$, we force the coefficient on the edge between $Z_j$ and $Z_k$ should be the same as the edge between $Z_j$ and $Z_l$, i.e., the edge connected to these binding nodes should be the same. After the merging stage, we get a complete graph with $m$ nodes.*

*Stage 2: Removing useless edges. Then for any $V$ and $V'$ if there is no edge between these two nodes, we set the coefficient of 'removing' relation as 1 and others as 0, which means we remove those edges from the logical rules. Finally, for $R_i(V_i, V_i')$ in the ground-truth logical rule, we set the coefficient of the correct relation on the corresponding edge as 1 and others as 0.*

*After all these manipulations, we can get that*

$$s_r(\{X_j\}, Y) = \sum\nolimits_{Z_1=1}^{|\mathcal{V}|} \cdots \sum\nolimits_{Z_K=1}^{|\mathcal{V}|} \prod\nolimits_{i=1}^{|\mathcal{E}_c|} A_i[V_i, V_i'], \quad (7)$$

*where*

$$A_i = \sum\nolimits_{k=1}^{|\mathcal{R}|} \left( \exp(\beta_{ik}) / \sum\nolimits_{k'=1}^{|\mathcal{R}|} \exp(\beta_{ik'}) \right) \bar{A}_k^+,$$

*becomes*

$$s_r(\{X_j\}, Y) = \sum\nolimits_{Z_1=1}^{|\mathcal{V}|} \cdots \sum\nolimits_{Z_K=1}^{|\mathcal{V}|} \prod\nolimits_{i=1}^n A_i[V_i, V_i'], \quad (8)$$

*where*

$$A_i = \sum_{k=1}^{|\mathcal{R}|+2} \left( \exp(\beta_{ik}) / \sum_{k'=1}^{|\mathcal{R}|+2} \exp(\beta_{ik'}) \right) \bar{A}_k,$$

*which means the Eq. (7) becomes the number of $(Z_1, \ldots, Z_K)$ that satisfies the logical rule. Note that, this is not the only way to construct the parameters to achieve this goal.*

## C  EXPERIMENTS

### C.1  DATASET

The statistics of five real-world datasets are reported in Table 6.

Table 6: Datasets statistics for learning complex logical rules.

|  | #Triplets | #Relations $|\mathcal{R}|$ | #Entities $|\mathcal{V}|$ |
|---|---|---|---|
| Kinship | 9587 | 25 | 104 |
| UMLS | 5960 | 46 | 135 |
| Family | 28356 | 12 | 3007 |
| Douban | 310 | 3 | 100 |
| Yelp | 283 | 3 | 100 |

### C.2  EVALUATION METRICS.

For each query $R_{\text{cpx}}(\{X_j\}, Y)$, we can calculate scores $s_r$ for all entities by our model. We sort the entities in a descending order of score and denote the rank of a right entity $Y$ as rank$(Y)$. Then, for each entity $Y$ we calculate MRR as follows,

$$MRR = \frac{1}{|\mathcal{D}_{\text{test}}|} \sum_{(\{X_j\}, Y) \in \mathcal{D}_{\text{test}}} \frac{1}{\text{rank}(Y)}$$

and Hit Rate at k as follows,

$$hit@k = \frac{1}{|\mathcal{D}_{\text{test}}|} \sum_{(\{X_j\}, Y) \in \mathcal{D}_{\text{test}}} \mathbb{I}\left[\text{rank}(Y) \leq k\right],$$

where $\mathbb{I}[\cdot]$ is an indicator function, which returns 1 if the statement is true and otherwise returns 0.

### C.3  LINK PREDICTION

We conduct the experiment of link prediction on three datasets Kinship, UMLS and Family following the setting in  Sadeghian et al. (2019). To be fair, we remove the embedding information in logical rule mining methods to purely compare the expressive ability about logic rules. We test all the methods for chain-like rules(Neural-LP (Yang et al., 2017), DRUM (Sadeghian et al., 2019) for more complex rule learning with only one input and one output, although they may learn inaccurate or wrong rules. We further add GraIL (Teru et al., 2020), the state-of-the-art method designed for inductive link prediction, as a baseline. As shown in Table 7, we can observe that our model achieves comparable performance compared with existing works.

Table 7: Performance comparison of link prediction on three real-world datasets.

| Method | Kinship | | | | UMLS | | | | Family | | | |
|---|---|---|---|---|---|---|---|---|---|---|---|---|
| | MRR | Hit Rate | | | MRR | Hit Rate | | | MRR | Hit Rate | | |
| | | @10 | @3 | @1 | | @10 | @3 | @1 | | @10 | @3 | @1 |
| DRUM | .52 | .83 | .60 | .37 | **.80** | **.92** | **.83** | **.73** | .71 | .90 | .84 | .58 |
| GraIL | **.55** | **.85** | **.63** | .41 | .72 | .91 | **.83** | .59 | **.85** | **.95** | **.92** | .76 |
| Ours | **.55** | .72 | .61 | **.44** | .70 | .79 | .72 | .65 | .82 | .94 | .84 | **.78** |

