# OpenReview forum: "Differentiable Learning of Graph-like Logical Rules from Knowledge Graphs"
_ICLR.cc/2021/Conference — Reject_

### Official Review · AnonReviewer2 · 2020-10-27
**Addressing a difficult problem with moderate success**

**Rating:** 5
**Confidence:** 4

**Review:**

This paper addresses the problem of logic structure learning, AKA, finding rules that generate a noisy knowledge graph. The innovation here is that the learning algorithm can address a much broader spectrum of rules than current methods. The key ingredient of the proposed method is a differentiable scoring function that models the rules.

* I think the authors are making the problem unreasonably complicated formalism to express the scoring function. It would have been much easier to express their algorithm in higher order tensors for equations (1),(2),(3)
* The A_i matrices they want to learn can be consolidated in a 3-dimensional tensor. Then what they need in order to obtain the s_r scoring function is a sumproduct over this tensor. In reality wha they are trying to learn is this huge but potentially sparse tensor. I would also like to point that from equation (3) it is implied that their algorithm cannot learn arbitrarily long recursive relations. Notice that in equation (3) each relation is used once only. For example, you cannot have A_3[Z1, Z2]A_3[Z2, Z3]....
* Another limitation which I think is important is that it does not learn the negation. That is probably an easy fix, but I think it is worth mentioning.

* The experiments are a bit limited in scope, but they are convincing. I realize that for the complexity of the algorithm it might be hard to scale to more datasets. I would have preferred if some timing results were reported too.

* What I find odd is that the authors didn’t try to model the 3-dimensional tensor A with a deep network. It would have definitely helped them scale better. I am not sure how they would have implemented the merge and remove operator but this is worth investigating

Overall I think the work is interesting and novel, but it is far from being practical for the moment.

---

> ### Author Response · Authors · 2020-11-25
> **Thank you for your valuable comments! We add real-world experiments for complex graph-like rules and dive into all your insightful suggestions one by one.**
>
> Thank you for your valuable comments! We very much appreciate it. Please allow us to respond to your questions one by one as follows.
>
> **Q1**: I think the authors are making the problem unreasonably complicated formalism to express the scoring function. It would have been much easier to express their algorithm in higher order tensors for equations (1),(2),(3)
>
> **Response**: Thank you for your insightful comments. Maybe our presentation misleads you. In our paper, we represent the score both in a scalar manner in Eq. (1) and in a tensor manner by Einsum in Eq. (5). Furthermore, in Sec 4.4, we show that for chain/tree-like rules, the score can be simply represented by matrix production or element-wise production, but for graph-like rules, we can only use Einsum to represent the score function instead of traditional computations like matrix production or element-wise production.
>
> **Q2**: The $A_i$ matrices they want to learn can be consolidated in a 3-dimensional tensor. Then what they need in order to obtain the $s_r$ scoring function is a sumproduct over this tensor. In reality what they are trying to learn is this huge but potentially sparse tensor. I would also like to point that from equation (3) it is implied that their algorithm cannot learn arbitrarily long recursive relations. Notice that in equation (3) each relation is used once only. For example, you cannot have $A_3[Z1, Z2]A_3[Z2, Z3]...$.
>
> **Response**: Thank you for your insightful comment. Based on your useful suggestion, we further analyze how to consider the negation relation based on our framework.
>
> **Q3**: Another limitation which I think is important is that it does not learn the negation. That is probably an easy fix, but I think it is worth mentioning.
>
> **Response**: Thank you for your insightful comment. Based on your useful suggestion, we further analyze how to learn the negation in our framework. To consider negation, we can construct the corresponding reverse relation for each original relation (replacing $1$ by $0$ and replacing $0$ by $1$).
>
> **Q4**: The experiments are a bit limited in scope, but they are convincing. I realize that for the complexity of the algorithm it might be hard to scale to more datasets. I would have preferred if some timing results were reported too.
>
> **Response**: Thank you for your valuable comments. Based on your valuable suggestion, we report the running times of our method and other baseline methods in Table 4. Furthermore, for making our experiment more convincing and showing our model is powerful for real-world applications, we conduct experiments of learning more complex graph-like rules on two more real-world datasets (Yelp and Douban).
>
> **Q5**: What I find odd is that the authors didn’t try to model the 3-dimensional tensor A with a deep network. It would have definitely helped them scale better. I am not sure how they would have implemented the merge and remove operator but this is worth investigating
>
> **Response**: Thank you for your insightful comment. We follow the Neural-LP and DRUM about how to tackle tensors. I think your mentioned idea of using deep networks to model the tensor is very novel and interesting, which we would investigate in future work.

---

### Official Review · AnonReviewer4 · 2020-10-28
**Compelling core idea, could be tightened up on clarity and rigor**

**Rating:** 4
**Confidence:** 3

**Review:**

#### Summary

This paper proposes a differentiable model to learn "graph-like" (ie, more general than tree-like) logical rules over a knowledge base from example instances of the target relation. The key contributions are the generalization beyond tree-like rules, an Einsum expression notation, and an experimental evaluation.

#### Strong and weak points

I found Figure 1 and the associated running examples to be extremely helpful. The decomposition and definition of the three tasks was also very clear. On the other hand, I found the notation and presentation text of Definition 1 and Section 4.1 to be dense and confusing. The auxiliary removing and merging relations could also have benefitted from some more explanation like a diagram or worked example.

The key innovation (in my reading) is the use of the learnable $\beta$ softmax parameters to determine which relations should be "active" in a given learned rule, as well as the normalized score objective which would penalize rules with many "spurious" groundings that satisfy the antecedent of the implication.

The significance or usefulness of graph-like rules could be established more convincingly. The Figure 1 running example seems a little contrived, and the experimental evaluation for these rules had to rely synthetic data. It would be helpful to identify important applications or research directions that would be unblocked by this work.

Experimental results against strong baselines show comparable performance on simpler (chain and tree-like) rules, and much stronger results (as expected) on graph-like rules.

The actual mechanics of what gets computed and how are not very clear. The einsum notation is nice, but it is not rigorously shown why or how the operations cannot be expressed as matrix operations, or what actual computational graph PyTorch would generate under the hood to run learning and inference.

#### Recommendation (accept or reject) with one or two key reasons

I would argue to reject primarily on clarity and rigor concerns, and somewhat less so on significance of graph-like rules.

#### Supporting arguments

I think there are some potentially exciting contributions here in terms of the beta-softmax formulation and being able to handle graph-like rules. That said the core technical content could be developed more clearly, and underlying details about complexity and computation should be handled more carefully. Some more specificity around use cases or research directions for graph-like rule learning could also further strengthen the submission.

#### Questions to clarify / additional evidence required

Why is the target relation called $R_{cpx}$ (ie, what does "cpx" mean)? This notation doesn't seem common in the related works.

"NP-hard problem" is mentioned a few times but without citation or clarification. The "Computational complexity" Big-O term is given without explanation or justification, and furthermore it was not clear what $m$ meant. Also, is the differentiable training approach guaranteed to find global optima, or only local ones? Is there an approximation factor or rounding gap between solutions found by this approach and the ground truth combinatorial optimum solution?

Section 4.1: minor typos "evaluate the plausibility of a logical _rule_" and "_adjacency_ matrix".


#### Additional feedback to improve

It might be worth reviewing the literature in logical rules learning, eg, [Turning 30: New Ideas in Inductive Logic Programming](https://arxiv.org/pdf/2002.11002.pdf) might be relevant. Relatedly, I think the terminology of "rule groundings" and related ideas from [Markov Logic Networks](https://homes.cs.washington.edu/~pedrod/papers/mlj05.pdf) would probably be helpful here. This work also seems particulary related to [Learning the Structure of Markov Logic Networks](https://homes.cs.washington.edu/~pedrod/papers/mlc05a.pdf).

It would be helpful to at least mention the normalized score earlier in the paper when the intuition for rule scoring is introduced, as I was immediately curious why this wouldn't lead to rules which add "easily satisfied" free variables and edges in order to increase the score.

---

> ### Author Response · Authors · 2020-11-25
> **Thank you for your valuable comments! We add real-world experiments for complex graph-like rules and analyse the related works in ILP and MLN.**
>
> Thank you for your valuable comments! We very much appreciate it. Please allow us to respond to your questions one by one as follows.
>
> **Q1**: The significance or usefulness of graph-like rules could be established more convincingly. The Figure 1 running example seems a little contrived, and the experimental evaluation for these rules had to rely synthetic data. It would be helpful to identify important applications or research directions that would be unblocked by this work.
>
> **Response**: thank you for your helpful suggestions. We have added two more real-world datasets (Yelp and Douban) for more complex graph-like rules to support our model's strength. Furthermore, we conduct a case study on Douban datasets to show that our model can mine graph-like rules on real-world datasets.
>
> **Q2**: The actual mechanics of what gets computed and how are not very clear. The einsum notation is nice, but it is not rigorously shown why or how the operations cannot be expressed as matrix operations, or what actual computational graph PyTorch would generate under the hood to run learning and inference.
>
> **Response**: thank you for your helpful suggestions. Based on your helpful suggestion, we add a brief introduction of Einsum in the main text and a detailed introduction of Einsum in the appendix. Einsum is a flexible convention of matrix calculation that unifies matrix multiplication, element-wise multiplication and some more complex matrix/tensor calculation that cannot be expressed by these ordinary operators (e.g., calculating the scores for graph-like rules).
>
> **Q3**: Why is the target relation called $R_{cpx}$ (ie, what does "cpx" mean)? This notation doesn't seem common in the related works.
>
> **Response**: Thank you for your insightful question. $R_{cpx}$ represent the relations that is equivalent to the complex compositional logical relations, where "cpx" is abbreviation for "complex".
>
> **Q4**: "NP-hard problem" is mentioned a few times but without citation or clarification. The "Computational complexity" Big-O term is given without explanation or justification, and furthermore, it was not clear what meant. Also, is the differentiable training approach guaranteed to find global optima, or only local ones? Is there an approximation factor or rounding gap between solutions found by this approach and the ground truth combinatorial optimum solution?
>
> **Response**: Thank you for your insightful questions. We have added supportive references and analyze the computational complexity in detail. Because the datasets are usually noisy, it is hard to provide a theoretical guarantee, but we will try this in future work. But as shown in Sec 5.3, we demonstrate that our model can learn accurate graph-like rules which is very close to the ground truth.
>
> **Q5**: It might be worth reviewing the literature in logical rules learning, eg, Turning 30: New Ideas in Inductive Logic Programming might be relevant. Relatedly, I think the terminology of "rule groundings" and related ideas from Markov Logic Networks would probably be helpful here. This work also seems particularly related to Learning the Structure of Markov Logic Networks.
>
> **Response**: Thank you for your helpful suggestions. We totally agree that inductive logic programming is very relevant to our work. Based on your suggestion, we add a new subsection (see Sec. 3.2) to thoroughly review the related works in this line, and particularly analyze the relationship between our work and Learning the Structure of Markov Logic Networks.
>
> **Q6**: It would be helpful to at least mention the normalized score earlier in the paper when the intuition for rule scoring is introduced, as I was immediately curious why this wouldn't lead to rules which add "easily satisfied" free variables and edges in order to increase the score.
>
> **Response**: Thank you for your helpful suggestions. We totally agree that mentioning the normalized score earlier is very important for understanding. Based on your valuable suggestion, we mention the normalized score when we first introduce the scoring function.

---

### Official Review · AnonReviewer1 · 2020-10-29
**Relevant and important topic, interesting idea with some novel twists**

**Rating:** 6
**Confidence:** 2

**Review:**

This paper proposes techniques that generate logical rules out of knowledge graphs; the idea is to produce more complex rules than usual by exploiting a differentiable formulation of the associated learning process. This is a relevant theme as rule learning from knowledge graphs is important in practice due to its potential interpretability (as compared to black-box schemes based on embeddings). The solution is relatively simple to describe, with a score that leads to differentiable learning, and some needed insights to obtain useful results. The empirical testing seems fine and does indicate that the method is useful in practice.

I must say that I was very interested in the "Einstein sum notation" but I could not quite see what it is and, more importantly, I could not see how this notation helps anything. Of course I understand that the authors do not have much space to explain details of this formalism, but some clarification would really help: I doubt that readers of ICLR papers will know this notation in detail. In fact, it would be nice to indicate how important this notation is in the whole system.

One question: looking at Figure 1 I see some questions on knowledge graphs and their possible answers, I cannot see exactly where are the "rules" there.

Overall the text can be followed but it could be revised so as to be made more didactic; also there are some annoying typos that should be fixed. For example:
- Abstract, line 6: "observing [that] the".
- Introduction, line 8: "ruleS".
- What is "widely-existed graph-like logical rules"?

---

> ### Author Response · Authors · 2020-11-25
> **Thank you for your valuable comments! We add more description about Einsum and add improve Fig. 1 to show the rules.**
>
> Thank you for your valuable comments! We very much appreciate it. Please allow us to respond to your questions one by one as follows.
>
> **Q1**: I must say that I was very interested in the "Einstein sum notation" but I could not quite see what it is and, more importantly, I could not see how this notation helps anything. Of course I understand that the authors do not have much space to explain details of this formalism, but some clarification would really help: I doubt that readers of ICLR papers will know this notation in detail. In fact, it would be nice to indicate how important this notation is in the whole system.
>
> **Response**: Thank you for your insightful comment. Based on your helpful suggestion, we add a brief introduction of Einsum in the main text and a detailed introduction of Einsum in the appendix. Einsum is a flexible convention of matrix calculation that unifies matrix multiplication, element-wise multiplication and some more complex matrix/tensor calculation that cannot be expressed by these ordinary operators (e.g., calculating the scores for graph-like rules).
>
> More specifically, it is a mathematical notational convention that is elegant and convenient for expressing summation and multiplication on vectors, matrices, and tensors.
> It can represent some calculation of multiple matrices that cannot be represented by matrix product or element-wise product.
> Specifically, it not only makes the expression of complex logic rules very simple and elegant but also make the implementation of this calculation simpler and more efficient.
> Here, we introduce the implicit mode Einsum because it can tackle more cases than the classical one, and this function has been implemented in many widely-used libraries such as Numpy and Pytorch. It takes a string representing the equation and a series of tensors as input and provides a tensor as output. The rules of Einsum are as follows,
>
> - the input string is comma-separated including a series of labels, where each term separated by commas represent a tensor (or matrix, vector) and each letter is corresponding to a dimension;
> - the terms in the string before '$\rightarrow$' means input variables, and the term after '$\rightarrow$' means output variables;
> - the same label in multiple inputs means element-wise multiplication on the dimensions of different input tensors;
> - the dimensions that occur in the inputs but not in the outputs mean that dimension will be summed, while others will remain in the output. When there is no dimension after '$\rightarrow$', it means all dimensions should be summed to get a scalar, and in such case '$\rightarrow$' can be omitted.
>
>
> **Q2**: looking at Figure 1 I see some questions on knowledge graphs and their possible answers, I cannot see exactly where are the "rules" there.
>
> **Response**: Thank you for your insightful comment. We revise the figure to make it more clear. Particularly, we explicitly add rules in the figure (see the third row in Fig. 1).
>
> **Q3**: Overall the text can be followed but it could be revised so as to be made more didactic; also there are some annoying typos that should be fixed.
>
> **Response**: Thank you for your detailed comment. We have thoroughly revised our paper, and we believe it would be substantially improved based on your valuable comments!

---

### Official Review · AnonReviewer3 · 2020-10-29
**Novel idea. Writing has much room for improvement. Experiments are synthetic**

**Rating:** 3
**Confidence:** 4

**Review:**

The paper aims to learn graph structured logical rules on knowledge graphs. It proposes three tasks, which I summarize as learning rule structure, instantiating a graph-structured rule (fill in a specific relation on the edge), and answering logical queries. However, many statements are entangled and mixed, the paper has many typos and inconsistency, making it hard to follow. I suggest the authors reformulate the tasks and the order or you can focus on one aspect/task and make it crystal clear.

Questions:
- What is the difference between logic rules and logical queries? The paper abuses terms like “logical rules”, “logical queries”, “logical rules for query”, making it extremely hard to follow. Are you learning logical rules or answering logical queries?
- According to the definition 1, figure 1 is not showing 3 logical rules but rather 3 logical formulas/queries, because there is no $R_{cpx}$.
- I don't quite follow and agree with the statement “Note that, $R_{comp}$ can both be a relation that exists in the KG and the human-defined logic rule for a query, which tends to be more complex” I think $\wedge_i^n R_i$ can be complex, no matter it is a chain, tree or graph, but $R_{cpx}$ should always be a concrete relation that exists on KG, otherwise it’s meaningless, because $R_{cpx}$ can simply represent anything. (I assume $R_{comp}$ is a typo and you mean $R_{cpx}$ here)
- Here is another confusing statement “Note that, in previous works (Hamilton et al., 2018; Ren et al., 2020), the logical rule $R_{cpx} = \wedge^n_i R_i$ is given”, according to your definition 1, shouldn’t the logical rule be $\wedge^n_i R_i \to R_{cpx}$?
- What is the relationship between $e(v_i, v_i’)$ and $R_i(v_i, v_i’)$?
- What is $G$ at the start of sec. 4.1, is it $G_e$ in section 2? There is also no $G$ anywhere in figure 1, you need to add some pointers.
- “We denote the adjacent matrix of relation $R_i$ as $A_i$ , which is a soft choice among adjacent matrices $A_k$ corresponding to all relations in R.” Then which one is the adjacency matrix $A$ or $A$ bar? According to your statement, it is $A$, but according to Eq. 2, it is $A$ bar.
- $\beta_{ik}$ comes out of nowhere. Why do you want to have this new parameter and what is the motivation?
- Why do you want to “by which we can use {$\beta_{ik}$} as learnable parameters to learn which relation should be assigned on each edge given the logical structure.”? I think you already have instantiated the edges right? E.g., in the right case in Figure 1, you already know that some edges are “read” and some edges are “friend”.
- The use of $S_r$ is not rigorous, for Eq. 3, it should be $s_r$({$X$},$Y)$ instead of $s_r(X,Y)$.
- Why do you have $A_1$ to $A_5$ in Eq.3, but in the example, you only have 3 relations: “read”, “read(inv)”, “friend”.
- In Sec. 4.3, it’s not “(1)”, “(2)”, but rather “Eq. (1)”, “Eq. (2)”
- Add reference and citations to the statement in the Computation Complexity paragraph.
- I did not find how the model answers logical queries (task 3) in the inference step, i.e., given a new $(${$X_j$}$,R_{cpx})$, how to find $Y$.
- In the case study section, it only shows how the model deals with task 1, i.e., when the logical structure is given. Considering there are 3 tasks, it would be better if you can show examples for the other 2 tasks.
- The datasets in the experiment section are small-scale and synthetic. What is the real-world application of the model? You claim that “For more complex rules, we generate a synthetic dataset because the real-world datasets are usually noisy.”, but isn’t the whole goal of the paper to model the real-world data and robust to noisy links?
- What is the evaluation setup? Table 2 lists the performance of query answering, (task 3), where are the results for the other 2 tasks? Also, I assume DRUM/Neural-LP can only handle link prediction in the original paper, there lacks details how you improve DRUM to model complex queries.

---

> ### Author Response · Authors · 2020-11-25
> **Thank you for your valuable comments! We add real-world experiments on complex graph-like rules and thoroughly revise our paper.  (1/2)**
>
> Thank you for your valuable comments! We very much appreciate it. Please allow us to respond to your questions one by one as follows.
>
> **Q1**: What is the difference between logic rules and logical queries? The paper abuses terms like “logical rules”, “logical queries”, “logical rules for query”, making it extremely hard to follow. Are you learning logical rules or answering logical queries?
>
> **Response**: Thank you for your insightful question. A logical query is usually defined based on compositional logical relations $\land_i^n R_i$. If we regard there is a relation $R_{cpx}$ between the input(s) and output of such a query, which can be either a relation in original KG or represented by a semantic sentence in Fig 1, then we can say there is a rule $\land_i^n R_i \rightarrow R_{cpx}$.
>
> **Q2**: According to the definition 1, figure 1 is not showing 3 logical rules but rather 3 logical formulas/queries, because there is no $R_{cpx}$.
>
> **Response**: Thank you for your insightful question. Based on your comment, we add $R_{cpx}$ in Fig. 1 to make it more clear (see the third row in Fig. 1).
>
> **Q3**: I don't quite follow and agree with the statement “Note that, $R_{comp}$ can both be a relation that exists in the KG and the human-defined logic rule for a query, which tends to be more complex” I think $\land_i^n R_i$ can be complex, no matter it is a chain, tree or graph, but $R_{cpx}$ should always be a concrete relation that exists on KG, otherwise it’s meaningless, because $R_{cpx}$ can simply represent anything. (I assume $R_{comp}$ is a typo and you mean $R_{cpx}$ here)
>
> **Response**: Thank you for your insightful question. You are right that $R_{cpx}$ should always be a concrete relation, which can be simply described by a set of pairs $(\{X_j\}, Y)$, and only the $\land_i^n R_i$. Here, we use the notation $R_{cpx}$ because its equivalent compositional logical relations $\land_i^n R_i$ is complex, e.g., graph-like rules.
>
> **Q4**: Here is another confusing statement “Note that, in previous works (Hamilton et al., 2018; Ren et al., 2020), the logical rule is given”, according to your definition 1, shouldn’t the logical rule be $\land_i^n R_i \rightarrow R_{cpx}$?
>
> **Response**: Thank you for your insightful question. In these works, they query $Y$ when $\land_i^n R_i$ and $\{X_j\}$ are given. As mentioned above, if we regard the relation of the input and output of these query as $R_{cpx}$, then it is explicitly given by the logical rule $\land_i^n R_i \rightarrow R_{cpx}$.
> But in our work, the logical rule $\land_i^n R_i \rightarrow R_{cpx}$ needs to be inferred from the input-output pairs $(\{X_j\}, Y)$ in the training stage (infer what kind of compositional logical relations is equivalent to the relation of the input-output pairs of such a query), and we need to infer what kind of $\land_i^n R_i$ could be equivalent to $R_{cpx}$ to support further query.
>
> **Q5**: What is the relationship between $e_(v_i,v_i')$ and $R_i(v_i,v_i')$?
>
> **Response**: Thank you for your insightful question. $e_(v_i,v_i')$ defines a edge in the rule but does not assign a concrete relation. If we assign a relaion on such an edge, then it becomes $R_i(v_i,v_i')$.
>
> **Q6**: What is $G$ at the start of sec. 4.1, is it $G_e$ in section 2? There is also no $G$ anywhere in figure 1, you need to add some pointers (see the second row in Fig. 1).
>
> **Response**: Thank you for your detailed comment. We have unified $G_e$ and $G$ as $G$ and added some pointers in the figure 1 to help understand.
>
> **Q7**: “We denote the adjacent matrix of relation $R_i$ as $A_i$, which is a soft choice among adjacent matrices corresponding to all relations in R.” Then which one is the adjacency matrix $A$ or $A$ bar? According to your statement, it is $A$, but according to Eq. 2, it is $A$ bar.
>
> **Response**: Thank you for your detailed comment. $R_i$ is corresponding to the relation on $e_i$, and $\mathcal{R}$ denotes all relations in KG. Based on your helpful comment, we change the sentence into "We denote the adjacency matrix of relation $R_i$ as ${A}_{i}$, which is a soft choice among adjacency matrices $\bar{A}_{k}$ corresponding to all relations $\mathcal{R}$ in KG".
>
> **Q8**: $\beta_{ik}$ comes out of nowhere. Why do you want to have this new parameter and what is the motivation?
>
> **Response**: Thank you for your detailed comment. $\{\beta_{ik}\}$ are a set of coefficient describing how to softly choose a relation for an edge in the logical structure $G$. Because we want to learn the logical rules in a differentiable manner, we need to relax such a combinatorial optimization into a continuous optimization problem by introducing a set of weights indicate which relation is assigned on a specific edge.

---

> > ### Author Response · Authors · 2020-11-25
> > **Thank you for your valuable comments! We add real-world experiments on complex graph-like rules and thoroughly revise our paper. (2/2)**
> >
> > **Q9**: Why do you want to “by which we can use $\{\beta_{ik}\}$ as learnable parameters to learn which relation should be assigned on each edge given the logical structure.”? I think you already have instantiated the edges right? E.g., in the right case in Figure 1, you already know that some edges are “read” and some edges are “friend”.
> >
> > **Response**: Thank you for your insightful question. In figure 1, we show some examples of logical rules, but in our problem, the logical rules need to be inferred. So in Sec 4.1, we know the structure but we do not know how to assign relations on each edge to form the logical rule, i.e., we do not know whether the relation on a specific edge is "friend" or "read". By using $\{\beta_{ik}\}$ as learnable parameters, if we learn a large weight of "read" on a specific edge bu maximizing the score, then it suggest that the relation on the edge are more likely to be "read".
> >
> > **Q10**: The use of $S_r$ is not rigorous, for Eq. 3, it should be $s_r(\{X\},Y)$ instead of $s_r(X,Y)$.
> >
> > **Response**: Thank you for your detailed comment. We have modified $s_r(X,Y)$ to $s_r(\{X\},Y)$ in Eq. 3.
> >
> > **Q11**: Why do you have $A_1$ to $A_5$ in Eq.3, but in the example, you only have 3 relations: “read”, “read(inv)”, “friend”.
> >
> > **Response**: Thank you for your detailed comment. $A_1$ to $A_5$ are 5 "undetermined" relations corresponding to 5 edges in the logical structure $G$, we need to infer what relation in KG should each of them be. For such a case, we need to learn $\{\beta_{ik}\}$, $i=1, \cdots n$ where $n=5$ is the number of edges in the structure and $k=1, \cdots |\mathcal{R}|$ where $|\mathcal{R}|=3$ is the number of relations in KG.
> >
> > **Q12**: In Sec. 4.3, it’s not “(1)”, “(2)”, but rather “Eq. (1)”, “Eq. (2)”
> >
> > **Response**: Thank you for your detailed comment. we have unified the notations of equations as Eq. (X).
> >
> > **Q13**: Add reference and citations to the statement in the Computation Complexity paragraph.
> >
> > **Response**: Thank you for your valuable suggestion. We added the reference to support the analysis of computation complexity.
> >
> > **Q14**: I did not find how the model answers logical queries (task 3) in the inference step, i.e., given a new $(\{X_j\},R_{cpx})$, how to find $Y$.
> >
> > **Response**: Thank you for your insightful question. Our description may mislead you. Actually, all $R_{cpx}$ should be seen in training datasets, so we can infer what kind of compositional relations $\land_i^n R$ is equivalent to $R_{cpx}$ and use $\land_i^n R$ to find $Y$ for $(\{X_j\}$ in testing datasets.
> >
> > **Q15**: In the case study section, it only shows how the model deals with task 1, i.e., when the logical structure is given. Considering there are 3 tasks, it would be better if you can show examples for the other 2 tasks.
> >
> > **Response**: Thank you for your valuable suggestions. Based on your suggestion to reformulate the problem, we have formulated the problem to mainly focus on task 3 and the other 2 tasks as auxiliary tasks or side products. Finally, we visualize the most insightful part of these experiments, the learned logical rules from the data to show the power of our model.
> >
> > **Q16**: The datasets in the experiment section are small-scale and synthetic. What is the real-world application of the model? You claim that “For more complex rules, we generate a synthetic dataset because the real-world datasets are usually noisy.”, but isn’t the whole goal of the paper to model the real-world data and robust to noisy links?
> >
> > **Response**: Thank you for your insightful questions. We have added two more real-world datasets (Yelp and Douban) for more complex graph-like rules to support our model's strength.
> >
> > **Q17**: What is the evaluation setup? Table 2 lists the performance of query answering, (task 3), where are the results for the other 2 tasks? Also, I assume DRUM/Neural-LP can only handle link prediction in the original paper, there lacks details how you improve DRUM to model complex queries.
> >
> > **Response**: Thank you for your detailed comment. We test all the methods for chain-like rules (Neural-LP, DRUM) for more complex rule learning with only one input and one output, although they may learn inaccurate or wrong rules.

---

### Author Response · Authors · 2020-11-25
**Response to all reviewers:**

We sincerely thank all the reviewers for their valuable comments! Here we briefly summarize our main revisions compared to the original version.

- Add real-world experiments for complex graph-like rules, running time results, case studies on real-world datasets.

- Discuss related works in inductive logic programming and Markov logic networks.

- Add more description of Einsum and explain why it can tackle graph-like rules.

- Add references about computational complexity.

- Reformulate the problem and improve the writing of this paper.

---

### Decision · Program_Chairs · 2021-01-07
**Final Decision**

**Decision:**

Reject

**Comment:**

While all reviewers see a lot of value in the paper, it cannot be accepted in its current form: too many issues with clarity. A more focused paper, with clear task and contributions is recommended.
The revisions and answers to reviewer questions are greatly appreciated and go a long way towards addressing these concerns for a future submission.